# Suprapatellar Tibial Nailing: Future or Fad?

**DOI:** 10.3390/jcm12051796

**Published:** 2023-02-23

**Authors:** Matthew Ciminero, Hannah Elsevier, Justin Solarczyk, Amir Matityahu

**Affiliations:** Department of Orthopaedic Surgery, Orthopaedic Trauma Institute, University of California San Francisco, San Francisco, CA 94110, USA

**Keywords:** suprapatellar, tibial nailing, tibia fractures, semiextended, tibia

## Abstract

Over the last hundred years, there have been significant advancements in the way the Orthopaedic community treats tibial fractures. More recently, the focus of Orthopaedic trauma surgeons has been comparing the different techniques of insertion for tibial nails, specifically suprapatellar (SPTN) versus infrapatellar. The existing literature is convincing that there does not appear to be any clinically significant differences between suprapatellar and infrapatellar tibial nailing, with some apparent benefits of SPTN. Based on the current body of literature and our personal experience with SPTN, we believe the suprapatellar tibial nail will become the future for most tibial nailing procedures, regardless of fracture pattern. We have seen evidence of improved alignment in both proximal and distal fracture patterns, decreased radiation exposure and operative time, relaxation of the deforming forces, ease of imaging, and static positioning of the leg, which would be helpful for the unassisted surgeon, as well as no difference in anterior knee pain or articular damage within the knee between the two techniques.

## 1. Introduction

Over the last hundred years, there have been significant advancements in the way the Orthopaedic community treats tibial fractures. In the early 1900s, the predominant treatment method was immobilization using splinting and casting, regardless of fracture displacement. It was not until the 1940s that Kuntscher’s nail [1] was introduced as a reliable method for treating long bone fractures; however, this too had limitations given the lack of interlocking screws. Since then, many changes have been made to intramedullary nails. Some examples include alterations in the material and size of the nails, the addition of multiple interlocking screw options to control for the rotation and shortening of length unstable fractures, and the introduction of a Herzog bend to decrease the insertional force required to implant the nail.

More recently, the focus of Orthopaedic trauma surgeons has been comparing the different techniques of insertion for tibial nails, specifically suprapatellar versus infrapatellar. It is well described that with infrapatellar nailing (IPTN) techniques, the knee needs to be hyperflexed to obtain the appropriate start site and trajectory. This leads to a procurvatum deformity due to the pull of the extensor mechanism on the proximal segment. In order to place a guidewire down the axis of the medullary canal while clearing the patella, increasing amounts of flexion need to be obtained. Ultimately, a posteriorly directed guidewire can lead to worsening of the procurvatum deformity upon nail insertion. Additionally, fluoroscopic imaging is burdened by requiring the C-arm to dramatically tilt in plane with the flexed knee. Lastly, the surgeon is required to work on stepstools or with their arms above their shoulders in order to ream in a superior to inferior direction. Due to all of these issues, there is a clear advantage to being able to insert tibial nails with the leg in a semiextended position. Krettek et al. developed techniques such as blocking screws or Poller screws to obviate these known issues while performing an infrapatellar nail [2]. Despite available means of successfully implanting an infrapatellar tibial nail with difficult proximal metaphyseal tibial fractures, new techniques were devised to help with these fracture patterns.

In 1996, Tornetta and Collins [3] presented their paper on the semiextended technique using a medial parapatellar arthrotomy to improve outcomes in proximal metaphyseal tibial fractures. The authors noted in their experience that about 15 degrees of displacement occurs in proximal third tibial shaft fractures when the knee is flexed 80–90 degrees. This is due to the displacing force of the quadriceps on the proximal segment. Proximal third tibial fractures are therefore the most affected by positioning and benefit the most from a semiextended approach. By keeping the leg in ~15–20 degrees of flexion, the surgeons could neutralize the force of the quadriceps and focus on the proper and safe entry point. The ideal starting site as described by Tornetta is the medial aspect of the lateral tibial spine in the coronal plane and at the articular margin on the sagittal image [4] [Figure 1]. It was not until 2010 that Eastman et al. showed in a cadaveric study that a tibial nail could be passed in a retropatellar fashion with a much smaller approach than the medial parapatellar concept proposed by Tornetta [5]. After some time, companies began developing tibial nails that would be inserted proximal to the patella via a small split in the quadriceps tendon and could be passed beneath the patella using sleeves to protect the patellofemoral joint, i.e., the suprapatellar tibial nail (SPTN). These sleeves are made of metal or plastic to protect the patellofemoral cartilage and vary depending on the company used.

## 2. Surgical Technique

The author’s preferred technique is to place the patient supine on a radiolucent bed. A hip bump is used on the operative extremity to keep the patella facing straight up. The ipsilateral upper extremity is taped over the chest to take tension off the brachial plexus in the bumped position, while the contralateral arm is 90 degrees to the body on an arm board (Figure 2). The operative extremity is elevated on a ramp, which can be a premade radiolucent foam ramp or one made from folded sheets. This allows for unobstructed lateral radiographs from the C-arm (Figure 3). Both the ramp and the non-operative leg are taped down to the bed. It is rare for the patellofemoral joint to be too tight to insert the trocar for the safe passage of the SPTN. However, it is still prudent for the surgeon to assess the patellar mobility preoperatively and scrutinize the injury films for signs of patellofemoral arthritis, both of which may lead the surgeon to choose a different type of nail. If there is significant tightness anticipated, an infrapatellar nail can be employed, or a retinacular release can be performed to increase patellar mobility. The limb is shaved, prepped, and draped in a standard sterile fashion. It is the author’s preference not to use a tourniquet unless a concomitant periarticular injury requires a larger approach. If needed, a small bolster of sterile towels can be used to ensure adequate knee flexion on the ramp for nail insertion. Similar towel bolsters of various sizes can be added under specific parts of the leg to aid in fracture reduction.

A 2-centimeter (cm) incision is then made approximately 1–2 cm proximal to the superior pole of the patella (Figure 4). The soft tissues are dissected with electrocautery until the quadriceps is encountered. The quadriceps tendon is identified and split sharply in the middle of the tendon through its full thickness. The retropatellar space is then manually palpated for the assessment of tightness, as well as to clear any adhesions or plica that may interfere with trocar passage. The patella can be manipulated superiorly, medially, or laterally, to assist in trocar insertion to avoid any retropatellar articular damage. The trocar is then inserted beneath the patella until the tibia is encountered. The surgeon should maintain distal pressure on the trocar handle to ensure the trocar does not inadvertently back out. Repeated insertion of the trocar can lead to patellofemoral damage. Certain SPTN systems allow the trocar to be pinned into either the tibia or femur to free up the surgeon’s hand. Once down on the tibia, the trocar can be removed from the cannula and replaced with a guidewire sleeve. The guidewire is inserted through the cannula, and the appropriate start site is confirmed using fluoroscopy (Figure 5). Generally, an upward force is required on the handle of the cannula in order to obtain the appropriate sagittal start site on the articular margin. This is best seen when the femoral condyles are perfectly overlapped. It should also be confirmed that the guidewire is on the medial aspect of the lateral tibial spine on coronal imaging. There will be significant changes in the apparent coronal start site due to parallax, and thus, a perfect AP image should be obtained and scrutinized before proceeding. This is performed by either 50% bisection of the fibular head by the lateral tibial plateau or “twin peaks” of the tibial spines [6].

An optimal start site is necessary for a successful surgery, especially when dealing with proximal third metaphyseal fractures (Figure 6). If a start site is close but not perfect, many of the implant systems allow for the insertion of a multi-hole cannula that rotates to make small adjustments in whichever plane necessary, as is being used in Figure 5. It is common knowledge that a medial start site leads to valgus, a lateral start site leads to varus, and a posterior-directed guidewire leads to procurvatum. Although an upward force on the cannula allows for the appropriate start point on the articular margin, the opposite is helpful when using the opening reamer to obtain an entry pathway in line with the sagittal axis of the tibia. If a tight patellofemoral joint is encountered where the cannula is obstructing an accurate start point, the guidewire can be safely inserted without the cannula, and then, the cannula inserted over the top of the wire once positioned. Once the ideal start site is obtained, if the trajectory is not perfect, the wire can first be advanced partially. Next, let the wire be consumed by the opening reamer to allow the reamer to make the adjustments for improved trajectory. The second necessity of treatment of fractures with intramedullary devices is that fracture reduction precedes reaming and nail insertion. Although isthmic fractures reduce well with nail insertion, the authors believe that regardless of fracture location, the fracture should be reduced prior to reaming and nail insertion. This can be achieved with external manipulation, traveling traction, percutaneous clamp application, or formal open reduction with or without plate and screw application. Once the surgery is complete, the authors generously irrigate the joint to remove any reaming debris that may cause post-operative irritation (Figure 7). The quadriceps tendon is closed with a heavy suture, and the skin closed in a standard layered fashion.

## 3. Literature

The existing literature is convincing that there does not appear to be any clinically significant differences between suprapatellar and infrapatellar tibial nailing, with some apparent benefits of SPTN; a review article performed by Franke et al. reached a similar conclusion [7]. While the authors believe that SPTN has become the gold standard for tibial nailing, surgeons and clinicians must always remain critical and sceptical of the body of evidence available. A paper published this past year evaluating the quality of the randomized controlled trials (RCTs) on the subject at hand found that the RCTs were considered poor by the Modified Coleman and CONSORT systems. In that same study, half the articles supported SPTN over IPTN, whereas the other half demonstrated no difference between the two techniques [8]. Although there is consistency among the evidence that is being published, caution should be taken with how the evidence is translated into practice, especially when new techniques are introduced.

We have discussed some of the benefits of SPTN already, such as the ease of insertion, the static maintenance of the leg position, the relaxation of the deforming forces, and the ease of obtaining fluoroscopic imaging [9,10,11]. Of these benefits, one in particular has been shown to affect radiographic outcomes. In 2016, Avilucea et al. demonstrated significantly decreased malalignment in distal third tibial shaft fractures when treated with an SPTN [12]. They concluded that the ability to maintain a steady leg was the likely cause of this outcome. We knew of the benefits of SPTN for proximal tibia fractures, but now, we have evidence it is also beneficial for distal third tibia fractures, which may have seemed less intuitive.

With the assistance of cadaveric studies by Tornetta et al. [4], we can minimize the damage to the articular structures when inserting tibial nails. As this was a concern for traditional IPTN techniques, it remains a concern for the newer SPTN. Fortunately, with the available evidence, there appears to be an even lower incidence of damage to the intra-articular structures with SPTN, as shown by Bible and colleagues [13]. Leary et al. also found there is no evidence of damage to the knee structures on both insertion and extraction for the SPTN in a cadaveric knee study [14]. Of greater concern and focus in the literature is the potential damage to the patellofemoral surfaces. Many papers have been published with no conclusive evidence that SPTN causes patellofemoral damage. When the nail was just coming out in 2010, Gelbke et al. found a statistically significant increase in mean patellofemoral contact pressures for SPTN (2.13 MPa); yet, it was not high enough to cause cartilage damage with a *single* impact (>25 MPa) or from sustained impact (4.5 Mpa) [15]. It is important to note that once the cannula is inserted in the retropatellar space, it is integral to not let it come out to avoid multiple reinsertions, which can cause clinically relevant cartilage damage and nullify the ostensibly low risk demonstrated in the above paper. This was followed in 2014 by Sanders et al., who looked at SPTN patients with a minimum 1-year follow up and assessed patient-reported outcome scores, MRI scans, as well as arthroscopic imaging (in a sub-group of patients) before and after nail insertion. They found no clinical complications or patellofemoral articular damage [16].

Another question commonly researched is whether patients receiving an SPTN have any significant anterior knee pain compared to IPTN patients. Despite these concerns, thus far, the literature has not shown any differences between the two nailing techniques for anterior knee pain or kneeling pain [17,18,19]. Serbest et al. also used arthroscopy in their study on 21 patients after each SPTN and found no association with either anterior knee pain or functional limitations of the knee at 1-year follow up [20]. Some studies have showed improved knee pain when compared with IPTN. In 2018, Chan et al. evaluated 12 RCTs and found SPTN reduced knee joint pain, visual analogue score, fluoroscopy time, and sagittal alignment. They also found better Harris hip score, Lysholm knee score, short-form 36 questionnaire, range of motion, and rates of “excellent” and “good” outcomes. There were no significant differences in operative time, blood loss, length of hospital stay, union time, and coronal alignment between groups [21]. Similar results were obtained in a small retrospective review of 20 SPTN versus 20 IPTN demonstrating decreased radiation exposure, OR time, superior outcome scores at 3 months for SPTN, and no observed complication differences [22].

The SPTN is not perfect though. A recent study evaluated 139 open tibial shaft fractures which were managed by an SPTN approach and found one case of septic arthritis following suprapatellar tibial nailing [23]. We also believe that in cases of severe patellofemoral arthritis, the SPTN may not be a good option given the retinacular releases that would likely be necessary to enable insertion of the instrumentation.

## 4. Conclusions

Based on the current body of literature and our personal experience with the SPTN, we believe the suprapatellar tibial nail will become the future for most tibial nailing procedures, regardless of fracture pattern. We have seen evidence of improved alignment in both proximal and distal fracture patterns, decreased radiation exposure and operative time, the ease of imaging, and the static positioning of the leg, which would be helpful for the unassisted surgeon, as well as no difference in anterior knee pain or articular damage within the knee between the two techniques. As previously mentioned though, the Orthopaedic community should be prudent with their evaluation of the literature and be judicious in appropriately evaluating the introduction of new technology into the Orthopaedic arena.

## Figures and Tables

**Figure 1 jcm-12-01796-f001:**
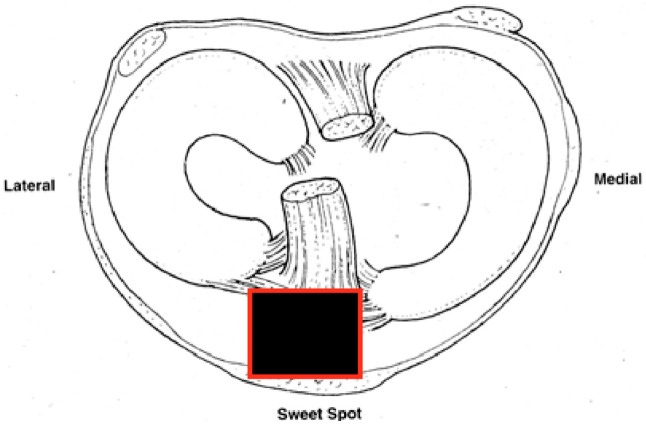
Axial cross section of knee with soft tissue attachments.

**Figure 2 jcm-12-01796-f002:**
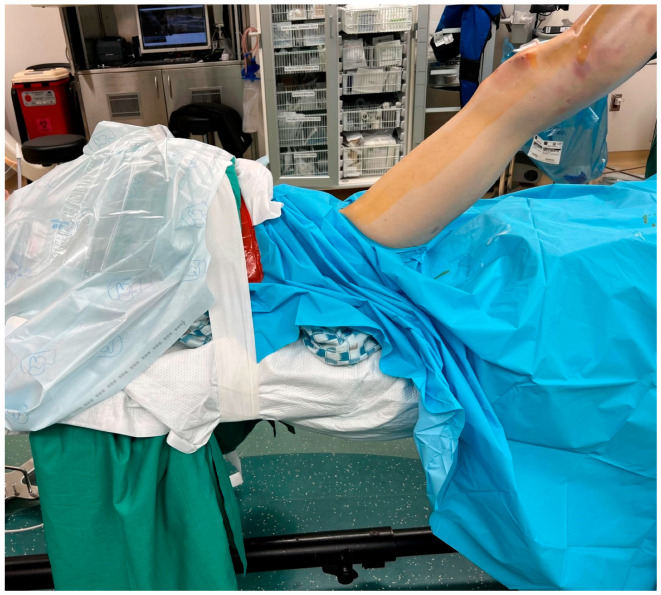
Pre-operative positioning with hip bump, arms taped over chest to reduce brachial plexus traction, bone foam ramp for unobstructed sagittal imaging.

**Figure 3 jcm-12-01796-f003:**
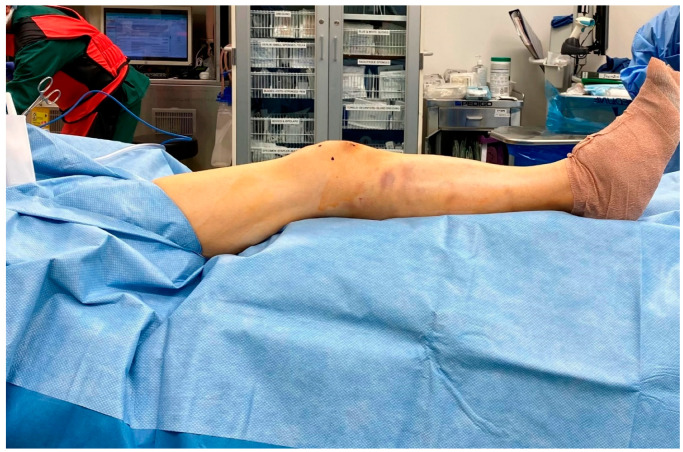
Surgical draping.

**Figure 4 jcm-12-01796-f004:**
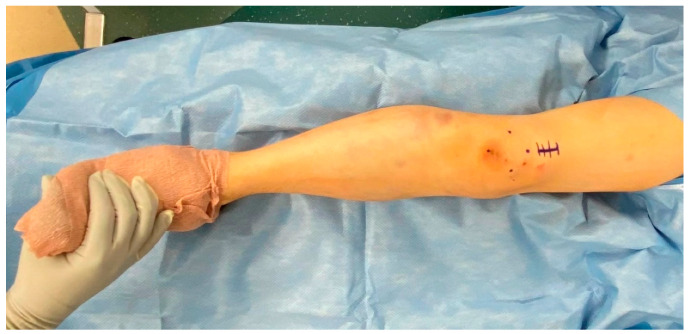
Patella marked out and incision planned one centimetre proximal to the proximal pole of the patella.

**Figure 5 jcm-12-01796-f005:**
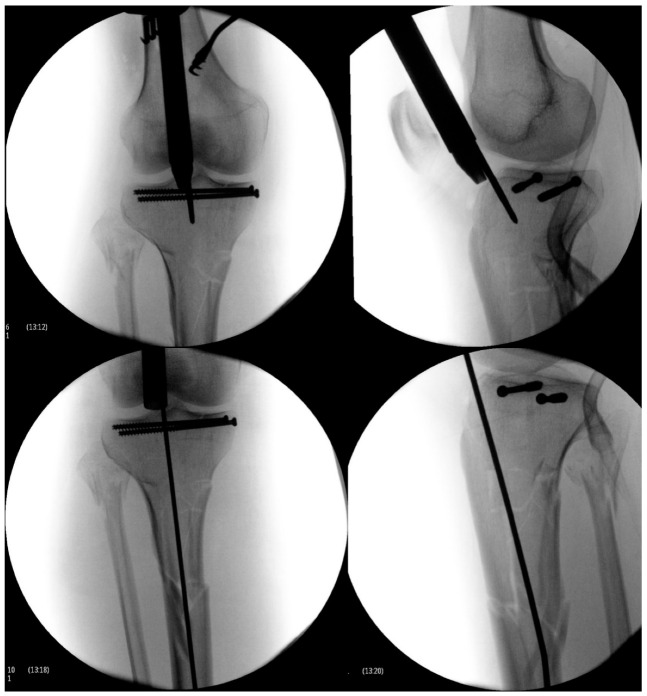
Start site illustrated on fluoroscopy for the tibial nail.

**Figure 6 jcm-12-01796-f006:**
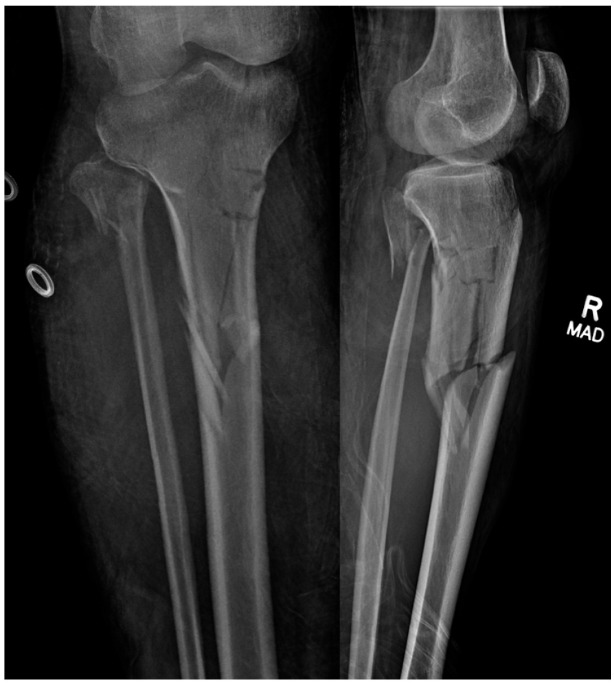
Pre-operative injury films.

**Figure 7 jcm-12-01796-f007:**
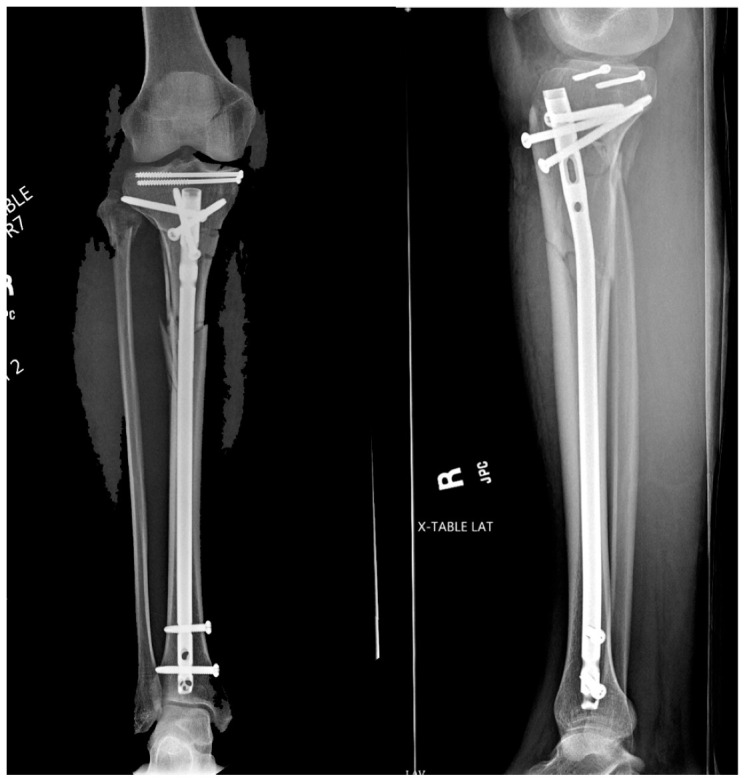
Immediate post-operative images.

## Data Availability

Data for this review article can be found in the references section.

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
