# Peer review of "Suprapatellar Tibial Nailing: Future or Fad?"

_jcm, 2023, doi:10.3390/jcm12051796_

Round 1
Reviewer 1 Report
The paper is dealing with a relevant topic and concludes absolutely right. However, the content of the paper is already well known. The description of the operative technique makes up a too large part of the paper. This is common sense. Which special type of cannulas are used to prevent cartilage from violation? Which different types of nails do you recommend in case of low patellar mobility or patellofemoral Arthritis (line 77)? Fracture reduction is recommended prior to reaming and the nailing procedure including opening up the fracture and plating. This is not necessary in SPTN because fracture reduction will happen automatically after positioning the patient in most cases.
The review of the literature as a basis of an argumentation in favor of SPTN is without any own scientific merit. Additionally, the review is not complete in relation to problems with implant removal via the suprapatellar approach and damage of intraarticular structures by choosing an entry point as recommended in figure 1 (Lig. transversum genus and meniscus).
Author Response
There are plastic and metal cannulas, this depends on the company. The literature doesn't distinguish between the two in terms of cartilage damage; it is just been proven via various imaging techniques to not have significant infrapatellar cartilage damage. I can add this to the paper.
If there is no patellar mobility or patellofemoral arthritis and joint space narrowing, then infrapatellar nailing can certainly still be employed, or a retinacular release to increase mobility.
I've never seen a displaced tibial shaft fracture reduce automatically from just being in the semi extended position.
I'm confused by the late statements. The literature as the basis of scientific merit is the only science available. What else do we use the literature for if not to aid in evidence based medicine?
I cannot find any papers looking at implant removal through a suprapatellar approach. I would have to respond anecdotally: if there is concern for infection as the reason for implant removal, I would remove it through an infrapatellar approach.
Reviewer 2 Report
Dear Authors, the topic is interesting but the work is incomplete. No data is reported, no control group. Also no sections of materials and methods and discussion are present. No objective is defined. The scientific article cannot be approved.
Author Response
It is a review article on a technique, not a scientific article.
Round 2
Reviewer 1 Report
Suprapatellar tibial nailing is an established technique, particular in proximal extraarticular fractures. The technique also has advantages in tibial shaft and distal extraarticular fractures because of easier radiological control of the procedure and less soft tissue stress compared to the infra patellar approach. All these advantages are already well known. Please compare: Franke J. et al. Suprapatellar nailing of tibial fractures - Indications and technique. Injury 47 (2016) 495-501.
Accordingly the present paper is a repetition of this technical note. The above named paper should be added to the references. The paper should be minor revisions, because the suprapatellar technique and the advantages should be more widely known in the community of trauma surgeons.
Author Response
I see the similarities and added the citation to include their study.
Reviewer 2 Report
Dear Authors the description is good notwithstanding the interest for reader is poor
Author Response
Understood.